https://doi.org/10.1038/s42004-022-00673-9　　OPEN
# Pre-arranged building block approach for the orthogonal synthesis of an unfolded tetrameric organic–inorganic phosphazane macrocycle

Ying Sim[1,6], Felix Leon[1,6], Gavin Hum[1], Si Jia Isabel Phang[1], How Chee Ong[1], Rakesh Ganguly[1,2], Jesús Díaz ![ORCID] [3✉], Jack K. Clegg ![ORCID] [4✉] & Felipe García ![ORCID] [1,5✉]

Inorganic macrocycles remain challenging synthetic targets due to the limited number of strategies reported for their syntheses. Among these species, large fully inorganic cyclodiphosphazane macrocycles have been experimentally and theoretically highlighted as promising candidates for supramolecular chemistry. In contrast, their hybrid organic–inorganic counterparts are lagging behind due to the lack of synthetic routes capable of controlling the size and topological arrangement (i.e., folded vs unfolded) of the target macrocycle, rendering the synthesis of differently sized macrocycles a tedious screening process. Herein, we report—as a proof-of-concept—the combination of pre-arranged building blocks and a two-step synthetic route to rationally enable access a large unfolded tetrameric macrocycle, which is not accessible via conventional synthetic strategies. The obtained macrocycle hybrid cyclodiphosphazane macrocycle, cis-[$\mu$-P($\mu$-N$^t$Bu)]$_2$($\mu$-p-OC$_6$H$_4$C(O)O)]$_4$[$\mu$-P($\mu$-N$^t$Bu)]$_2$ (4), displays an unfolded open-face cavity area of 110.1 Å$^2$. Preliminary theoretical host–guest studies with the dication [MeNC$_5$H$_4$]$_2^{2+}$ suggest compound 4 as a viable candidate for the synthesis of hybrid proto-rotaxanes species based on phosphazane building blocks.

[1] Division of Chemistry and Biological Chemistry. School of Physical and Mathematical Sciences, Nanyang Technological University, 21 Nanyang Link, Singapore 637371, Singapore. [2] Shiv Nadar University, NH-91, Tehsil Dadri, Gautam Buddha Nagar, Uttar Pradesh 201314, India. [3] Departamento de Química Orgánica e Inorgánica, Facultad de Veterinaria Universidad de Extremadura, Cáceres, Spain. [4] School of Chemistry and Molecular Biosciences, Cooper Road, The University of Queensland, St Lucia 4072 QLD, Australia. [5] Departamento de Química Orgánica e Inorgánica, Facultad de Química, Universidad de Oviedo, Julián Claveria 8, Oviedo 33006 Asturias, Spain. [6] These authors contributed equally: Ying Sim, Felix Leon. ✉email: jdal@unex.es; j.clegg@uq.edu.au; garciafelipe@uniovi.es

Organic macrocycles have been attractive synthetic targets due to their numerous applications[1–4]. In contrast, many inorganic macrocycles remain chemical curiosities due to the lack of well-established synthetic approaches capable of rationally selecting their size, topological arrangement and conformation.

Among the families of inorganic compounds capable of forming a broad range of cyclic (and acyclic) frameworks are cyclophosphazane species based on group 15 (G15) $P_2N_2$ building blocks, which have been widely explored due to their chemical versatility[5]. These species have shown to be excellent neutral and/or anionic ligands for metal coordination[6–13] and building blocks for the construction of larger molecules, as well as utility in biological applications and supramolecular chemistry[14–23].

Great chemical versatility combined with the fact that many frameworks across the main group elements are isoelectronic and isostructural makes the cyclophosphazane family an ideal model system for synthetic route benchmarking and proof-of-concept studies (Fig. 1)[24–28]. In addition, many synthetic routes and conceptual developments can be extrapolated to other families comprising different main group element combinations.

In this context, cyclodiphosphazane-based inorganic macrocycles have drawn attention due to both their promise in host–guest chemistry and anion sensing[22,23,29–36]. However, their potential to play an important role in supramolecular chemistry is limited by the lack of large macrocycle scaffolds and adequate synthetic routes to access them. The general synthetic approach is the reaction of bifunctional linkers (both organic and inorganic) with dichlorocyclodiphosphazane, $[ClP(\mu\text{-}N^tBu)]_2$ (1), which generate the targeted macrocycles in moderate to good yields.

Hybrid $P_2N_2$ macrocycles have attracted particular interest over the past two decades due to their dual inorganic–organic nature. Generally, these species are obtained via a one-step route using an equimolar amount of reactants (i.e., 1 and the organic linker) in the presence of excess triethylamine[37–39]. The size of the macrocyclic products formed (e.g., di-, tri- and tetrameric species), however, is strictly determined by the nature of the bifunctional linkers employed[37,38,40].

Using this approach, a large number of examples of condensation reactions comprising symmetric bifunctional organic linkers (L–L) to produce hybrid inorganic–organic macrocycles of general formula $[(\mu\text{-L-L})P(\mu\text{-}N^tBu)]_n$ have been reported[37–42]. Among these, the majority are dimeric species ($n = 2$)[39,40,42,43], with only two trimeric ($n = 3$) and two tetrameric ($n = 4$) macrocycles ever isolated and fully characterised[37,38,40,41]. Notably, tetrameric macrocycles ($n = 4$) have only been obtained in a folded topological conformation from 1,4-diaminobenzene in moderate yields[41], and with resorcinol as a minor product—reported in 2005 and 2012, respectively[37].

One would expect that reactions involving asymmetric bifunctional organic linkers (L–L') would provide a certain degree of size control over the $P^{III}_2N_2$ macrocyclic outcome due to the differential chemical properties between the inequivalent terminal moieties. However, these species have been exclusively isolated as small dimeric $P^{III}_2N_2$ species of the type $[(\mu\text{-L–L'})P(\mu\text{-}N^tBu)]_2$ displaying both cis and trans arrangements[44,45].

Despite the tremendous efforts invested over the last two decades in the synthesis of these species, no synthetic routes capable of controlling the size of hybrid $P^{III}_2N_2$ macrocycles have been reported to date. Hence, the development of alternative methodologies that allow rational control over the reaction to the end $P^{III}_2N_2$ macrocyclic product is key to the advancement of the field of phosphazanes and main group chemistry at large.

Since traditional one-step equimolar approaches have repeatedly proven inadequate to access larger frameworks and/or enable control over macrocycle size (and topological configuration), we postulated that the differential properties present in asymmetric linkers, in combination with elaborate orthogonal and multi-step synthetic routes, can be capitalised on to obtain species otherwise inaccessible via conventional synthetic routes. Orthogonal synthetic methodologies have been widely applied to organometallic, organic, polymeric and supramolecular chemistry[46–48]. However, orthogonality in main group synthetic chemistry is still in its infancy[49–51].

As a proof-of-concept, we herein describe the selective synthesis of the largest reported hybrid inorganic–organic $P^{III}_2N_2$ macrocycle via the synergic combination of pre-arranged $P^{III}_2N_2$ building blocks, and an elaborated two-step synthetic route. This approach is in stark contrast to the previously described traditional single-step synthetic route, which only affords small dimeric macrocyclic species (Fig. 2).

In addition, preliminary DFT theoretical studies suggest this macrocycle as a good candidate for the synthesis of hybrid protorotaxanes species.

## Results and discussion

**Bifunctional organic linker requisites and linker selection.** Initially, we reasoned that the differential chemical and reactivity properties present in asymmetric linkers could be leveraged on for the synthesis of a cis-pre-arranged building block, which in turn would grant access to large cis hybrid inorganic–organic cyclodiphosphazane macrocycles comprising asymmetric linkers.

For this purpose, we hypothesised that an ideal linker would require: (i) the presence of two functionalities with pronounced $pK_a$ differences, (ii) a rigid backbone with an appropriate substitution pattern, and (iii) a strong preference for an exo,exo conformation upon reaction with $P_2N_2$ (to enforce unfolded topologies).

After an in-depth review of the literature and exhaustive analysis of the previously published work on $P^{III}_2N_2$ macrocycles, we selected 4-hydroxybenzoic acid as the optimal organic linker for the following reasons (see Fig. 3).

Firstly, this compound compromises two acidic moieties (i.e., –COOH and –OH with $pK_a$s ~4.20 and ~10, respectively) bonded to a rigid phenyl group in a para conformation—prerequisites (i) and (ii), respectively. In addition, all previously reported cyclodiphosphazane species containing –O(O)C–X (X = Ph–O–, Ph–CN, –C(O)O–, and $CF_3$) moieties directly bonded to the $P^{III}_2N_2$ ring exhibit a strong preference for an exo,exo conformation[44,52]—prerequisite (iii)—which is in contrast with commonly reported hydroxyl and amino groups (Fig. 3a)[45,53–55].

Secondly, in the previously reported $P^{III}_2N_2^{exo,exo}$ monomer containing two –OC(O)PhCN substituents, the distances between the distal para substituents (ca. 7 Å) is greater than the average distance between substituents directly attached to the $P^{III}_2N_2$ ring (ca. 4 Å) (Fig. 3b). This topological conformation creates a

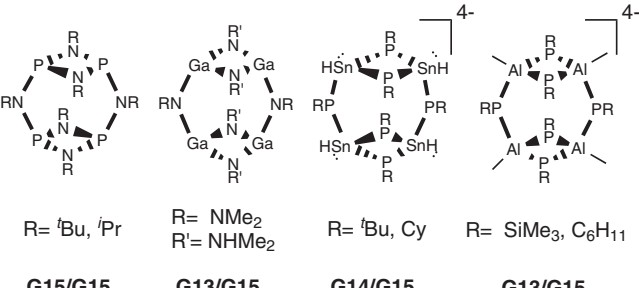

**Fig. 1 Isostructural main group frameworks.** Examples of isoelectronic and/or isostructural main group fully inorganic dimeric macrocycles.

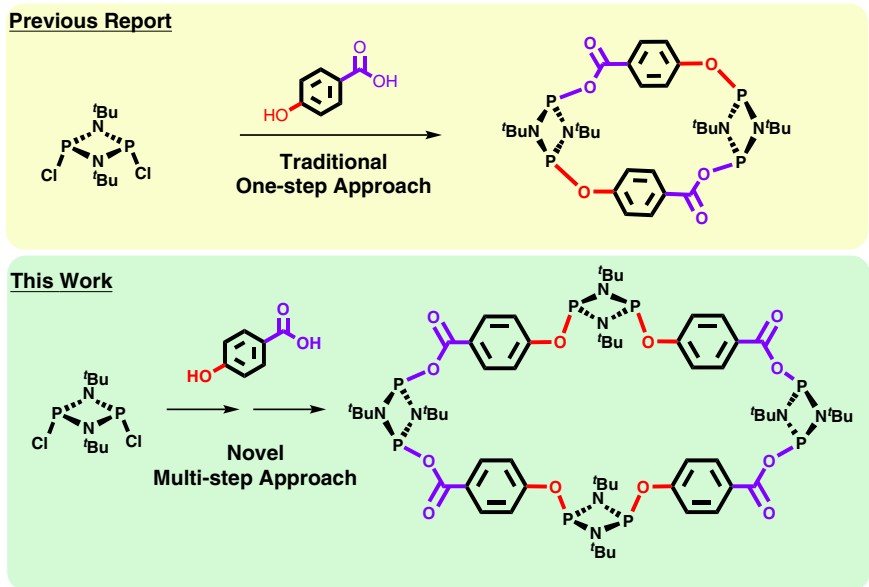

**Fig. 2 Schematic comparison of the presented work with respect to conventional synthetic routes.** Traditional one-step synthetic route leading to dimeric species (top)[44], and the herein reported two-step synthetic strategy to afford an unfolded tetrameric macrocycle using pre-arranged building blocks (bottom).

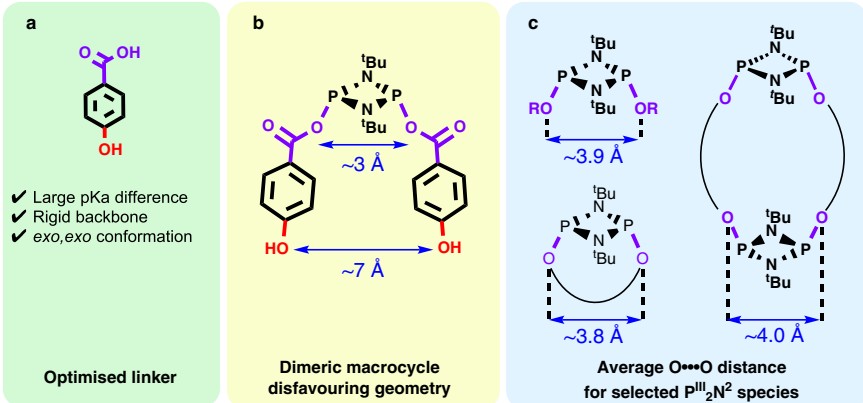

**Fig. 3 Bifunctional linker selection criteria. a** Linker prerequisites, (**b**) dimeric $P_2N_2$ macrocycle disfavouring geometry building block, and (**c**) previously reported O···O distances on previously reported cyclic and acyclic $P^{III}_2N_2$ species (see Supplementary Table S4 in Supplementary Discussion 1).

geometrical mismatch that we envisage will disfavour the formation of small dimeric $P^{III}_2N_2$ macrocycles (Fig. 3c shows the average O···O distance for cyclic and acyclic $P^{III}_2N_2$ species containing organic substituents or linkers).

Lastly, this linker has been previously reported to produce a trans dimeric macrocycle, trans-[(µ-O-C$_6$H$_4$ -COO)P(µ-N$^t$Bu)]$_2$ (**2a**), as the sole product when the conventional single-step equimolar synthetic route was used (Fig. 4)[44]. This selectivity for dimeric macrocycles will serve as a base model and benchmark for our proof-of-concept approach (Fig. 4, synthethic route to trans dimeric macrocycle **3a**).

**Synthesis of *cis* pre-arranged building block**. With this in mind, we set out to synthesise the targeted $P^{III}_2N_2$ building block comprising two –OC(O)PhOH substituents. Compound **1** (Supplementary Fig. S1) was reacted with two equivalences of 4-hydroxybenzoic acid in THF at −78 °C in the presence of a base (i.e., Et$_3$N), and the resulting mixture was left to gradually warm up to room temperature and further stirred for 3 h (Supplementary Fig. S2).

The in situ $^{31}$P-{$^1$H} NMR spectra—shown in Fig. 5, **top**—revealed a singlet resonance signal at δ 172.5 ppm, which was attributed to the acyclic cis di-substituted monomeric building block, [(HOC$_6$H$_4$(O)CO)P(µ-N$^t$Bu)]$_2$ (**2a**, in Fig. 4) (cf. previously reported [(CN)C$_6$H$_4$(O)CO)P(µ-N$^t$Bu)]$_2$ counterpart δ 173.0 ppm)[44].

Furthermore, there were no resonances recorded consistent with the formation of the aryl oxide cis di-substituted derivative [(HOC(O)C$_6$H$_4$O)P(µ-N$^t$Bu)]$_2$ (**2b**, in Fig. 4), since no signals were observed at δ ~144 ppm—where cis acyclic alkoxide or aryloxide species would have been expected[56–58]. Additionally, no signals for the trans di-substituted (2c, in Fig. 4) monomer (i.e., mixed aryl oxide and alkoxide), and negligible amounts of the previously reported trans dimeric macrocycle (3a, Fig. 4 left) were observed (Supplementary Fig. S2).

Unfortunately, attempts to isolate analytically pure samples of **2a** were unsuccessful. Hence, the cis pre-arranged building block **2a** was used in situ throughout our studies.

**Synthesis of all cis hybrid $P^{III}_2N_2$ tetrameric macrocycle**. After the successful formation of the targeted *cis* pre-arranged building

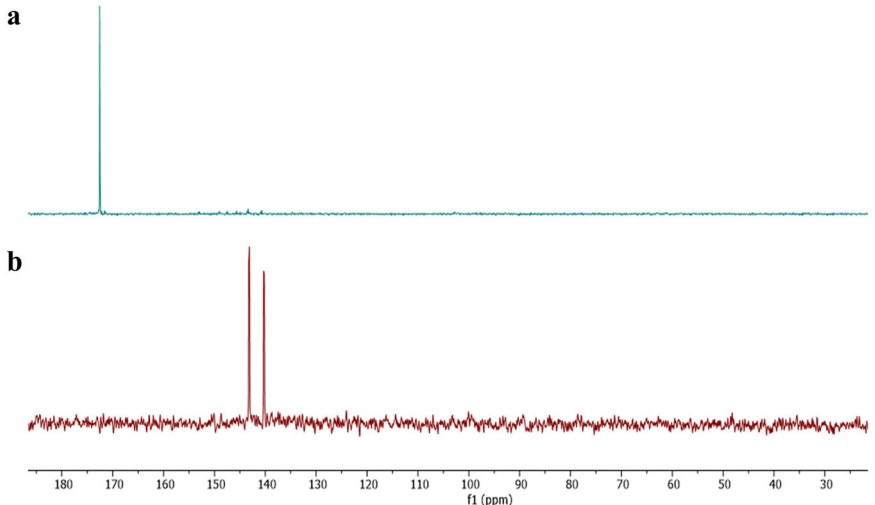

**Fig. 4 Proposed reaction pathway towards dimeric and tetrameric macrocycles.** Conventional synthetic approach (left) and two-step methodology to macrocycle **4** using pre-arranged building blocks (right).

**Fig. 5 Comparative in situ $^{31}$P-{$^1$H} NMR spectra.** Conventional one-step approach reaction leading to **3a** (**b**) and step 1 the two-step synthetic route leading to **2a** (**a**).

block, **2a** was subsequently reacted with compound **1** in the presence of Et$_3$N at −78 °C, and left to stir overnight (Fig. **4, right**). The in situ $^{31}$P{$^1$H} NMR spectra recorded revealed two singlet resonance signals at δ 181.3 and 152.4 ppm along with small resonance signals corresponding to the dimeric *trans* macrocycle **3a**, presumably due to fragmentation and

rearrangement products of **2a** or its derivatives (for $^{31}$P{$^1$H}, $^1$H and $^{13}$C NMR spectra of **4** see Supplementary figs. S2–S5, respectively)[42,59]. Despite these minor side products, pure compound **4** was isolated by crystallisation of a concentrated solution in hexanes (32% yield). SC-XRD studies indicated the successful formation of an all-cis tetrameric macrocycle obtained, *cis*-[μ-

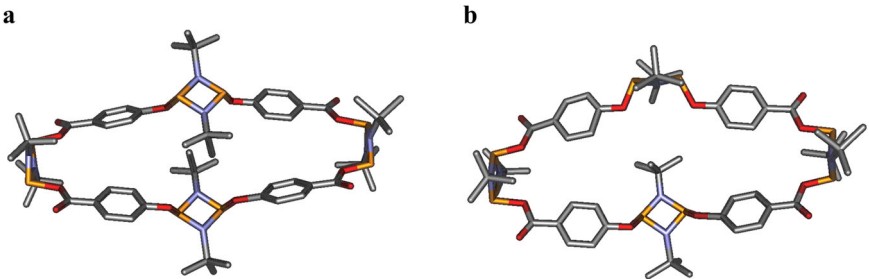

**Fig. 6 Solid state structure of 4.** Two different conformers (panels **a**, **b**) of cis tetrameric macrocycle (**4**) depicted as stick model. Hydrogen atoms and disorder have been omitted for clarity. CIF file can be found in Supplementary Data file 2.

P($\mu$-N$^t$Bu)]$_2$($\mu$-$p$-OC$_6$H$_4$C(O)O)]$_4$[$\mu$-P($\mu$-N$^t$Bu)]$_2$ (**4**), which is in stark contrast to the dimeric macrocycle obtained using conventional one-step equimolar synthetic routes (see Fig. 6).

Notably, since the selected linker imposes an exo,exo conformation around the -C(O)O- moiety, compound **4** features an unfolded topology which leads to the largest cavity among the hybrid inorganic–organic cyclodiphosphazane macrocyclic family to date. This uniquely large ellipsoidal cavity displays two orthogonal axes of 19.36 Å and 7.24 Å, and an area of 110.1 Å$^2$, which are significantly larger than any previously reported hybrid P$^{III}_2$N$_2$-macrocycles (Supplementary Table S1). Moreover, this large cavity enables partial rotation of central P$^{III}_2$N$_2$ bonded to hydroxyl groups, which is illustrated by the presence of two different macrocyclic conformers within the crystal lattice, resulting in significant disorder in the diffraction data collected (Fig. 6). In one conformer, the two opposite P$^{III}_2$N$_2$ units are parallel to each other, while in the other conformer, these P$_2$N$_2$ units are almost perpendicular (Supplementary Fig. S10). To the best of our knowledge, the observed ability of the central P$^{III}_2$N$_2$ units to partially rotate has never been observed in any cyclodiphosphazane macrocyclic species reported.

The relative mobility of the central P$_2$N$_2$ units around the hydroxyl axis in **4** is further confirmed by $^1$H NMR spectroscopy, where all the tert-butyl groups display a singlet resonance at δ 1.39 ppm and which split into two different environments upon cooling at 5 °C (Supplementary Fig. S7).

**Computational studies**. Notably, the O•••O bond distances in **4** provide geometrical cues to rationalise the observed preference for the formation of the tetramer **4** instead of the dimer **3b** (both with *cis* conformation). This suggests that the two free hydroxyl groups present in the cis pre-formed building block **2a** are separated by too large a distance (≈6.0 Å), disabling the cyclic arrangement closure (leading to **3b**) with one dichlorophosphazane unit and in turn resulting in the formation of the tetrameric macrocycle **4** (see Fig. 4).

The large **d$_o$** distance proposed—a critical structural feature of **2a**—is supported by the previously reported [((CN)C$_6$H$_4$(O)CO)P($\mu$-N$^t$Bu)]$_2$[44]. In this compound, the average separation between the –CN substituents on the distal para positions is approximately 7 Å. This separation is much larger than the distance between the two chlorine atoms in the dichloro substituted starting material **1** (≈4.1 Å)[60], or any other reported dichlorocyclodiphosphazanes (ranging from 3.8 to 4.7 Å) (Supplementary Table S3)[38,55,61–65]. Therefore, the subsequent addition of **1** resulted in the formation of tetrameric macrocycle **4** via condensation of two molecules of intermediate ***Int I*** (see Fig. 4)—instead of the conventionally obtained trans dimeric macrocycle **3a**.

Computational calculations were performed to both rationalise the experimental observations and gain insights into the mechanism involved in the formation of **4**. The energies of all the species and intermediates in Fig. 4 were calculated using

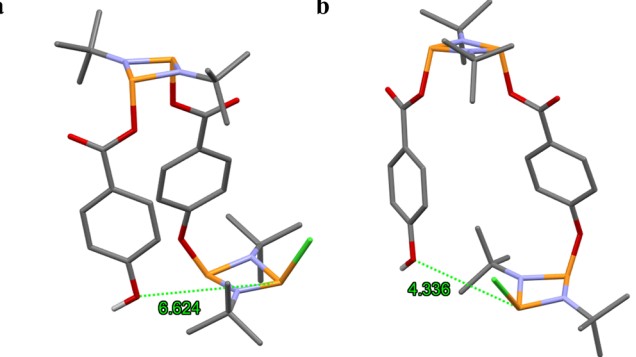

**Fig. 7 Proposed intermediate in the formation of 4.** Calculated Cl-**P•••O**H distance (in Å) at the minimum energy computed structure (**a**) and at the closest calculated distance via a Ph–O dihedral rotation (**b**).

Density Functional Theory and normalised including the corresponding starting materials and by-products (e.g., Et$_3$N or Et$_3$NH$^+$Cl$^-$) Attempts to obtain a complete energy profile of the reaction were unsuccessful, and thus only the relative ground estate energies of the different species involved were used throughout our discussion (See Supplementary Fig. S11–S14 and Supplementary Tables S5 and S6).

Firstly, the key role of the pre-arranged rigid monomer was investigated. In terms of thermodynamic stability, all three di-substituted (cis-**2a**, cis-**2b**, and trans-**2c**) where very close in energy (see supplementary Fig. S11). The most stable isomer of the three is where the organic linkers are attached to the phosphazane unit *via* the acid group (cis-**2a**, in Fig. 4). This observation is consistent with the experimental observations, where only the cis-**2a** isomer is observed in the in-situ $^{31}$P{$^1$H} NMR spectra, due to the higher pKa of the acid group.

Once **2a** is formed, it reacts with a **1** to form an asymmetrically substituted monomeric intermediate, [(HOC$_6$H$_4$(O)CO)–(P($\mu$-NtBu))$_2$–O(CO)C6H4O-(P($\mu$-NtBu))2-Cl] (***Int I***), where a new P–O bond has been formed. Once the first P–O bond has been formed, there are two possible reaction pathways (see Fig. 4 and Supplementary Fig. S11): (i) an intramolecular nucleophilic attack between the OH moiety and the terminal P-Cl within ***Int I*** to form a second P–O bond, and hence yielding the cis-dimeric macrocycle **3b** (not observed experimentally) (Pathway I in Supplementary Fig. S11), or (ii) an intermolecular reaction between two ***Int I*** molecules to produce the observed cis-tetrameric macrocycle **4** (Pathway II in Supplementary Fig. S11).

Within the context of these two possible reaction pathways, ***Int I*** was optimised, and its most stable configuration displays a Cl-**P•••O**H distance of 6.6 Å, indicating a difficult and unlikely nucleophilic attack to form **3b** (see Fig. 7a). Additionally, a 360° scan of the Ph–O–P dihedral angle was carried out to find the closest possible Cl–P•••OH distance, which was found to be 4.3 Å

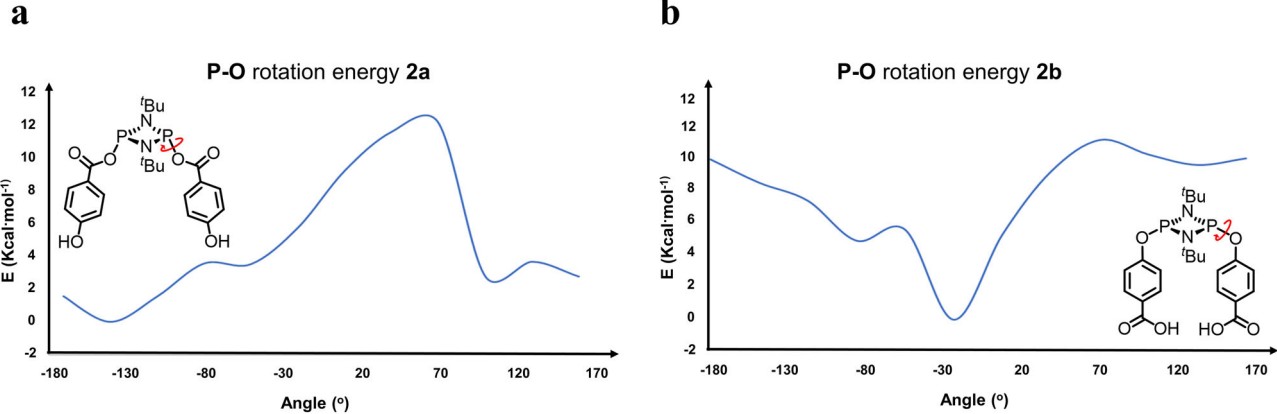

**Fig. 8 Calculated rotational energy barriers for monomeric *cis* isomers.** Scan of the rotation electronic energy along one of the **P–O** dihedral angles in **2a** and **2b**, (**a**,**b**), respectively.

(Fig. 7b). This distance is more than two-fold the average P–O bond distance observed in cyclodiphosphazane species, which further proves the difficulty of obtaining the cis-dimeric macrocycle **3b** using the pre-arranged building block **2a**[38,55,61–65].

Alternatively, rotation around the **P—O** bond in ***Int I*** could potentially place the P–Cl and –Ph–OH terminal moieties close enough in distance for the intramolecular nucleophilic attack required to form **3b** to take place. Within this context, an energy scan for the rotation around the **P—O**C(O)PhOH bond in **2a** was calculated, although the TS for this process could not be found, the scan energy profile provides with an estimation of the energy barrier for the process (see Fig. 8a). The calculated rotation displays a high energy barrier (~12 kcal•mol$^{-1}$), which is in good accord with the experimentally observed preference for exo,exo configuration displayed by all the previously reported cyclodiphosphazane species comprising –p–OC(O)–R moieties. In contrast, the **P—O** rotation barrier for then alkoxy derivative **2b** (i.e., the **P—O**PhC(O)OH bond) is roughly half (~6 kcal•mol$^{-1}$), which is consistent for the exo,endo preference observed this type of species (see Fig. 8b)[56–58].

Additionally, the cis- and trans- dimeric macrocycles (**3a** and **3b**, respectively) were optimised to gain insights on their relative stabilities. This could help rationalise both the observation of the previously reported trans dimeric macrocycle **3a** as a side product, as well as the failure to observe its cis-dimeric counterpart **3b** on thermodynamic grounds. Our DFT calculations indicate that **3a** is relative more stable than **3b**, which is aligned with the experimentally observed exclusive formation of **3a** from the 1:1 reaction of **1** with *p*-HOC(O)PhOH, presumably via the condensation of two ***Int II*** molecules (see Fig. 2 and Supplementary fig. S11—via conventional synthetic routes). Interestingly, the formation of a small amount of **3a** is observed in the in situ $^{31}$P NMR using our approach. This phenomenon can be attributed to a Cl$^-$ mediated cleavage of one of the **P–O** bonds in **2a**, leading to the formation of ***Int II*** and the subsequent formation of **3a**, which has been previously described for strained phosphazane species[59].

Finally, we computed the energies of both cis and trans isomers of the tetrameric macrocycle **4**. The computational calculations indicate that the cis isomer is thermodynamically favoured over its trans counterpart (i.e., 5.59 kcal•mol$^{-1}$, see Supplementary Fig. S14 and Table S6).

Overall, our theoretical studies suggest that both the strong preference for an exo,exo conformation of the –C(O)O– moieties combined with the stepwise synthetic route used are critical for

the successful formation of the tetrameric macrocycle herein reported.

**Theoretical assessment of the host–guest properties of 4.** While host–guest studies can be readily performed with robust and air-stable organic counterparts, the timeline for cyclophosphazane species generally spans much longer due to their air- and moisture-sensitive nature. For instance, the first fully inorganic NH-bridged pentameric macrocycle[35] was first reported in 2002[35], however, its first implementation[49] as a host-for small organic molecules was only experimentally demonstrated in 2019[20]. However, despite the longer implementation times, cyclophosphazane species have shown to versatile building blocks in supramolecular chemistry, capable of outperforming traditional organic building blocks[17–23].

Previous density functional theory (DFT) studies on the host–guest ability of cyclic and acyclic cyclodiphosphazane species have shown good agreement between theoretical and experimental values[36,66], which will provide a faster and reliable preliminary assessment of the potential of **4** as supramolecular building block.

Given the large cavity observed, which is comparable to other organic supramolecular building blocks used to synthesise rotaxane species[66,67], we performed DFT calculations to assess the host capacity of **4** with the dicationic 1,1'-dimethyl-[4,4'-bipyridine]-1,1'-diium (**5**) as a guest. Compound **5** has been commonly used to produce a wide range of archetypal [2] rotaxanes and higher-order multicomponent rotaxanes[67].

Our theoretical studies indicate that when **5** is used as a guest molecule, the host–guest complex is stabilised by 53.2 kcal•mol$^{-1}$ with respect to the free host and guest (Supplementary Fig. S15 and Table S7). This high stabilisation energy clearly shows a significant affinity of the host for the model guest[68–70], and it is comparable to the stabilisation provided when **21-crown-7** is used as a host (Supplementary Fig. S17 and Table S8). Notably, despite **4** having only four oxygen atoms, its stabilisation energy is only 3.8 kcal•mol$^{-1}$ lower than that of the **21-Crown-7**. In addition, **18-Crown-6** was also studied; however, the formation of the host–guest adduct is not favourable. The comparatively high affinity of P$^{III}_2$N$_2$ oxo species with respect to crown ethers has already been also theoretically demonstrated for an all-inorganic hexameric analogue[66].

In addition, the study of the electrostatic potential (ESP) surface for both **4** and its host–guest adduct with **5** provides further insights into the observed affinity. The ESP map of **4** shows a highly localised negative region around the four central oxygen atoms region of the macrocycle (Fig. 9, left). In contrast,

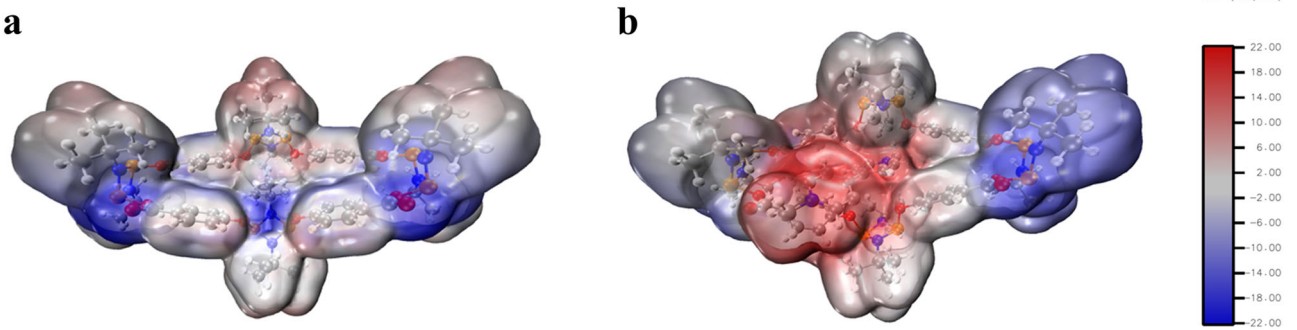

**Fig. 9 Electrostatic Surface Potential (ESP) maps.** ESP map for **4** (**a**) and ESP map for the pseudo-rotaxane host–guest adduct with **5** (**b**). ESP maps have been plotted at 0.03 isovalue.

upon formation of the pseudo-rotaxane host–guest adduct with **5**, there is a drastic change in the ESP map—displaying a highly positively charged central macrocycle area, and an enhanced negative character at the distal macrocyclic regions.

The presence of attractive host–guest interactions within the pseudo-rotaxane adduct formed between **4** and **5** is further supported by non-covalent interaction (NCI) analyses which display strong, attractive interactions between the central bridging oxygen atoms and the delocalised positive charge of the guest (Supplementary Fig. S16).

Overall, the large cavity within **4**, combined with the calculated favourable affinities to **5** offers an exciting opportunity to use **4** as a platform to produce hybrid rotaxane species based on cyclodiphosphazane species.

Currently, we are focusing our efforts on the synthesis of compound **4** counterparts with enhanced hydrolytic stability[44,50,56,71,72] to fully capitalise on the promising host–guest properties predicted in our studies.

## Conclusions

A two-step synthetic route combined with rationally pre-arranged building blocks has been demonstrated to obtain a unique tetrameric hybrid inorganic–organic macrocycle (**4**). This compound is not accessible through one-step conventional synthetic routes, which only produce small dimeric species.

In contrast to the standard exo,endo folded configuration of hybrid cyclodiphosphazane macrocycles, the herein reported tetrameric macrocycle displays a unique exo,exo unfolded topology leading to the largest cavity area of its kind (110.1 Å²).

Theoretical DFT host–guest calculations show a high affinity towards 1,1'-dimethyl-[4,4'-bipyridine]-1,1'-diium—a common building block in rotaxane chemistry—suggesting **4** as a good candidate to synthesise hybrid proto-rotaxanes species.

Finally, our work serves as proof-of-concept to highlight the need to invest greater efforts in developing a wide range of complex pre-arranged main group building blocks, which we envision will play a key role in developing novel supramolecular chemistry based on main group species.

## Methods

**Synthetic procedures.** Step 1 to obtain compound **2a**: A 25 mL THF solution of 1 (0.550 g, 2.0 mmol) and triethylamine (0.56 cm3, 4.0 mmol) was first prepared. To a 25 mL THF solution of p-hydroxybenzoic acid (0.552 g, 4.0 mmol), the pre-prepared THF solution was added dropwise at −78 °C. The reaction mixture was left to gradually warm to room temperature and left to stir for 3 h. According to the in situ ³¹P-{1H} NMR spectroscopy recorded, the reaction exclusively resulted in the formation of acyclic disubstituted cyclodiphosph(III)azane (compound **2a**), as indicated by the singlet resonance signal at approximately δ 172.54 ppm (Supplementary Fig. S2). This reaction solution was then used for step 2 without further workup.

Step 2 to obtain the unfolded tetrameric macrocycle 4: Another batch of 25 mL THF solution of 1 (0.550 g, 2.0 mmol) and triethylamine (0.56 cm3, 4.0 mmol) was added dropwise at −78 °C into the previous reaction mixture. Upon complete addition, the reaction mixture was stirred overnight. Solvent was removed under reduced pressure and hexane was added and the suspension filtered in celite. The filtrate was subsequently concentrated and left to crystallize at 5 °C. The product was collected as colorless crystals. Yield: 0.436 g (32%).

**Characterisation of Compounds.** For characterisation of compounds see Supplementary Methods and Supplementary Discussion 1–6. For NMR Spectra Supplementary Figs. S1–7 in Supplementary Discussion 1–3. For HRMS and IR spectra see Supplementary Figs. S8–9 in Supplementary Discussion 4. For SCXRD studies see Supplementary Fig. S10 in Supplementary Discussion 5 and Supplementary Data 2 for the cif file of compound **4**.

**Computational details.** For computational details see Supplementary Discussion 6 and for atomic coordinates see Supplementary Data 1.

## Data availability

The authors declare tha the data supporting the findings of this study are available within the paper and its supplementary information files. Atomic coordinates can be found in supplementary Data file 1. The X-ray crystallographic coordinates for structures reported in this Article, can be found in Supplementary Data file 2, have also been deposited at the Cambridge Crystallographic Data Centre (CCDC), under deposition number 1996702 for compounds **4**, and. These data can be obtained free of charge from The Cambridge Crystallographic Data Centre via www.ccdc.cam.ac.uk/data_request/cif.

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

## Acknowledgements

F. G. would like to thank A*STAR AME IRG (A1783c0003 and A2083c0050), MOE AcRF Tier 1 (M4011709) and NTU start-up grant (M4080552) for financial support. F. G. also thanks the support of Fundación para el Fomento en Asturias de la Investigación Científica Aplicada y la Tecnología (FICYT) through the Margarita Salas Senior Program (AYUD/2021/59709). F. L. would like to thank A*STAR for fellowship. J. K. C. acknowledge the support of the Australian Research Council through DP1901012036. J. D. thanks COMPUTAEX for granting access to LUSITANIA supercomputing facilities.

## Author contributions

F.G. conceived the research, obtained the funding for the project and jointly supervised the work. Y.S. and F.L. designed the experiments and performed the synthetic and characterisation experiments assisted by G. H., S. J. I. P, and H. C. O. R.G. and J. K. Clegg collected and solved the crystal data for compound **4**. F.L. and J.D. performed the theoretical studies. All authors analysed the data and participated in drafting and revising the manuscript.

## Competing interests

The authors declare no competing interests.
