## [Peer Review File · Communications Chemistry]

Reviewers' comments:

Reviewer #1 (Remarks to the Author):

The manuscript from García, Clegg, Díaz and col. describes the preparation of a new macrocycle composed of phosphazane building blocks. The manuscript is well written, and the results could be of interest for a wide audience, particularly after considering the future prospects of these kind of systems. While only "2-D" structures are treated here, it is not difficult to imagine the possibilities of these kind of systems with other spacers capable of introducing new groups for expanding the possibilities, either functional (catalysis, molecular recognition, rotaxanes, etc) or structural (supramolecular polymers, MOF, etc). Therefore, I consider that the manuscript should be published in Nature communications and that this interest justifies the urgency of its publication. However, before I would like that some minor problems with the manuscript could be polished.

The approach of using two groups with different pKa is very clever, and should be put into context with other methodologies for orthogonal functionalization (such as protection-deprotection, etc), highlighting its advantages.

The yield of the reaction is moderate (33%). Although small yields are common in the formation of organic macrocycles, one wonders whether this could be improved by selecting other reaction conditions. A possible competitive reaction is the formation of head-tail polymers, a reaction that is very sensitive to the concentration of the reactants. Also, template effects could favour the formation of the macrocycle, and the selectivity of the reaction could be affected by the solvent, temperature, etc. With such a large number of variables it is very unlikely that the optimal synthesis conditions were obtained after only one shoot. Although synthesis optimization could be out of the scope of a communication, including in the SI section the results from the different synthetic methodology tested (and not only the successful results) may help the readers of the communication to envisage the optimization procedure for the synthesis.

The DFT calculations helps to understand and justify some of the experimental results, but in some cases there is no agreement between both. The calculations have been performed only on starting materials, products and intermediates, and no transition state structure has been calculated or used in the discussion. I find this approach reasonable, considering that calculating the TSs could be problematic due to the different catalytic possibilities (surely Et₃N plays a role as a catalyst in the mechanism, but, for example, it is not easy to know a priori how many Et₃N molecules are present). However, it is important to be aware of the limitations of an approach in which the TSs are not considered at all... basically, the theoretical results could only match experimental results under thermodynamic control conditions, and it may not be the case (indeed, it is almost certainly not the case) for these reactions.

Hence, it is not possible to affirm, based on the Gibbs free energies of 2b, 2a and 2c, that the calculations support the experimental observation that only product 2a is obtained. The very similar energies of 2b, 2a and 2c would imply that a mixture of the 3 compounds will be obtained under thermodynamic control. Therefore, the only valid conclusion of the calculations is that this reaction is kinetically (not thermodynamically) controlled. The structures to the right of Figure 7 are also misleading. Compound 4-trans cannot be expected from Int I. Overall, probably an energy diagram like the one shown in Figure 7 is not the more appropriate way of showing these results, since one would expect to find in this diagram the TSs and barrier heights that could justify the reaction results

under kinetic control.

The absolute numbers of the energies of compound are also affected by the choice of the reference system. Since the reactions takes place in the presence of Et₃N, it is very likely than carboxylic acid deprotonates and forms an ionic pair with the protonated amine. Therefore, the zero-energy should correspond to this ionic pair. Solvent effects, that will be crucial in this case, would imply the use of a solvent model, and the translational entropy, that is modelled in QM packages using a gas-phase model, will also include a considerable error. Indeed, the numbers found are too large to be realistic (≈ 60 kcal/mol is similar to the heat of combustion of H₂!). However, luckily these numbers are not useful in the discussion, so I can save the effort of questioning their validity. My suggestion would be to get rid off these numbers in the discussion. In any case, they must not contain more than a single decimal figure.

However, calculations are useful to justify, on the basis of the geometries of the intermediates and rotation barriers, the preference for the formation of 4 cis over 3b (at least qualitatively). In my opinion, the rotation results included in the SI section should replace the energetic diagram and the discussion should focus more on these rotational barriers than in the energies of the products and intermediates.

The calculations are also useful to stablish the way of association of the phosphazane macrocycle with diamines. But in this case what I miss is the experimental observation of this complex. Ideally, the complex would crystalize and the structure would be observed by X-ray diffraction experiments, or more sophisticated NMR experiments (NOESY, DOSY, etc) could confirm the association. But even a simple NMR experiment would reveal displacement of the signals of the macrocycle upon the addition of the diamine, to confirm the theoretical expectations.

Reviewer #2 (Remarks to the Author):

This paper reports a new two-stage synthesis of an unfolded inorganic/organic cyclodiphosphazane macrocycle, which has been isolated and structurally characterized. The new precursors were carefully chosen to overcome the previous limitations of dimerization and allows isolation of a larger tetramer. The design features were clearly explained. Theoretical investigations provided a reasonable explanation to the observed selectivity of the macrocycle in terms of energy. Although this paper offers strong contributions in the synthetic aspect (however, reporting only one fully characterized compound), my concern is that the host-guest properties of the featured product 4 are only theoretical, which limited the impact of this paper. I have the following questions for the authors to address before I can consider recommendation for publication:

1. The capability (“outperforming traditional organic building blocks”) of most of the cyclophosphazane as building blocks cited in this paper (ref 20, 17-23) utilizes a N-H moiety for hydrogen bonding. Does the synthetic strategy offered by this paper apply to N-H derivatives? Alternatively, will the tert-butyl derivative offer superior guest-host properties in a similar way or any other way?

2. Page 10 Line 180: 32% of Compound 4 was obtained from the first batch of crystals. What is the

total isolated yield of 4? If it's not over 60-70% at least, this reaction is not as selective as it was claimed.

3. Page 11 Line 199-200: Is there any implications of this unprecedented rotation of the central P(III)₂N₂ units on its host-guest properties?

4. How is the thermal stability and solubility of the compound 4? The authors mentioned the air- and moisture sensitive nature which prevents the host-guest studies. What about simple NMR experiments? If compound 4 is not thermally stable, what does it decompose into?

5. For a synthesis-focused paper, there is only one fully characterized compound (4). Although NMR assignment has been shown in the current version of SI, I don't see experimental evidence for the reported assignment. The 2D NMR evidence for the assignment should be added in the SI. In figure S5, is the carbon signal assigned to 9 because it showed coupling to P?

Reviewer #3 (Remarks to the Author):

In this work, the innovative approach which is alternative to already conventional synthetic route has been developed, allowing the tetrameric macrocycles. This approach has an importance for supramolecular chemistry field in terms of the control of macrocyclic size. The paper which is written by Garcia et al. is original and acceptable for publishing in Communications Chemistry after minor revisions have been implemented. Some recommendations are given for authors to enhance the quality of the paper below:

1. The energy levels of compound 2a and 2b are very close to each other to synthesize compound 4 (cis). In addition, the difference between their values and compound 2c is nearly 0.49 kcal/mol. So, both compound 2a and 2b are possible candidate for the synthesis of compound 4. Only the chemical shift value in ³¹P NMR spectrum cannot be distinctive method. The reason why compound 2a or 2b is starting compound for the synthesis of compound 4 must be clarified.

2. The ³¹P NMR spectrum of the reaction mixture in which newly synthesized compound 4a were obtained, should be given in the paper. It was mentioned that compound 4 was obtained in a yield of 32%. What about the remaining part (68%)? Do the new compounds form or compound 2a remains without reacting? This should be clarified.

3. In Supplementary file, "first batch of crystals" was written for compound 4 in the experimental section. Why does it mention as first batch of crystals???

4. Is it possible to get different result if different base (for ex. NaH) rather than triethylamine was used? Why triethylamine was preferred?

5. There are some mistakes in the paper:

Page 10 line 168: It was said trans dimeric macrocycle (3a) was not observed but in Figure S2, it is seen it formed even little amount and marked with symbol at about 140 and 150 ppm.

Page 10 line 176: 'Resonance signals at 180 and 150.1 ppm are not compatible with the data in Supplementary file (Figure S3). The values should be corrected.

Page 10 line 178: The in situ ³¹P{¹H} NMR spectra recorded revealed two singlet 175 resonance signals at δ 180.0 and 150.1 ppm along with small resonance signals corresponding to the dimeric trans macrocycle 3a, presumably due to fragmentation and rearrangement products of 2a or its derivatives (see Supplementary figures S2-S6)... Figure S5 is related to ¹³C NMR spectrum of compound 4, so this is not relevant.

6. It is seen the R factors are high (R₁, wR₂ (>2σ(I)) 0.1331, 0.3575 R₁, wR₂ (all data) 0.1588, 0.3854), which means there is a problem in crystal quality but I/σ (I): 20.6 Rint factor below 10%. Why does this contradiction arise? The crystal data were collected at 100 K, but I see they were collected at

296 K / 23 °C. Low temperature data collection may prevent rotation of the phenyl ring and tert-butyl group. The data should be collected again at low temperature.

7. The capped stick model is preferred as the crystal displaying in both the paper and the supplementary part. ORTEP drawings should be given by showing the atom-labelling scheme. The bond parameter is included in Figure S10, but these parameters create uncertainty since the atoms were not labelled.

8. You need to update your cif. file by labelling the atoms correctly. The alert A and B errors on the checkcif file should be explained.

Reviewer #1 (Remarks to the Author):

The manuscript from García, Clegg, Díaz and col. describes the preparation of a new macrocycle composed of phosphazane building blocks. The manuscript is well written, and the results could be of interest for a wide audience, particularly after considering the future prospects of these kind of systems. While only “2-D” structures are treated here, it is not difficult to imagine the possibilities of these kind of systems with other spacers capable of introducing new groups for expanding the possibilities, either functional (catalysis, molecular recognition, rotaxanes, etc) or structural (supramolecular polymers, MOF, etc). Therefore, I consider that the manuscript should be published in Nature communications and that this interest justifies the urgency of its publication. However, before I would like that some minor problems with the manuscript could be polished.

The approach of using two groups with different pKa is very clever, and should be put into context with other methodologies for orthogonal functionalization (such as protection-deprotection, etc), highlighting its advantages.

ANSWER – As suggested by Reviewer we have include references and mentions to orthogonality in synthetic chemistry and main group chemistry. “we postulated that the differential properties present in asymmetric linkers, in combination with elaborate orthogonal and multi-step synthetic routes, can be capitalised on to obtain species otherwise inaccessible via conventional synthetic routes. Orthogonal synthetic methodologies have been widely applied to organometallic, organic, polymeric and supramolecular chemistry.^{46–48} However, orthogonality in main group synthetic chemistry is still in its infancy.^{49–51”} – **Highlighted in yellow (page 5)**

The yield of the reaction is moderate (33%). Although small yields are common in the formation of organic macrocycles, one wonders whether this could be improved by selecting other reaction conditions. A possible competitive reaction is the formation of head-tail polymers, a reaction that is very sensitive to the concentration of the reactants.

Also, template effects could favour the formation of the macrocycle, and the selectivity of the reaction could be affected by the solvent, temperature, etc. With such a large number of variables it is very unlikely that the optimal synthesis conditions were obtained after only one shoot. Although synthesis optimization could be out of the scope of a communication, including in the SI section the results from the different synthetic methodology tested (and not only the successful results) may help the readers of the communication to envisage the optimization procedure for the synthesis.

ANSWER – As pointed out by the reviewer, the manuscript highlights the successful results, but not the multiple failed attempts. We agree that the challenges faced during the optimization of the reaction will be invaluable to the reader as well, and thus we have included additional information in the SI section.

For step 1, to obtain the pre-arranged compound **2a**, the reaction is relatively simple with little complications. However, it should be noted that compound **2a** is not stable at room temperature if excess triethylamine is added, as it generates dimer **3a** over prolonged stirring. This indicates that the deprotonated hydroxyl groups are sufficiently nucleophilic to attack the phosphorus atom to eliminate hydroxybenzoic acid. As such, the treatment of **2a** with triethylamine should be kept at low temperature for only a short period of time.

For step 2, the reaction is sensitive to the presence of acid/base, temperature, and concentrations. Firstly, the addition of a base is essential for the reaction. When **2a** was added with **1** (at -78°C or -94°C prior to the addition of triethylamine), white fumes of HCl was immediately generated. In situ NMR at this point reveals that the majority of **2a** is likely to have undergone an acid-catalysed Arbuzhov rearrangement (see **Figure A1 below**).

Figure A1. $^{31}\text{P}\{^1\text{H}\}$ in situ NMR from the sequential addition of **1** followed by triethylamine.

However, as previously mentioned, basic conditions promoted the dimerization of **2a** or rearrangement of the linker at room temperatures. Therefore, the reaction was carried out at -78°C in the presence of triethylamine, in an effort to both reduce rearrangement and dimerization.

Concentration dependence of the reaction was also observed, as higher concentrations resulted in a broad and complex signal at around 170 ppm. This is likely to be longer chain oligomers due to head-tail polymers as pointed out by the reviewer. (**Figure A2**) Thus, low concentrations should be used to favour the formation of the tetramer over oligomers.

Figure A2. $^{31}\text{P}\{^1\text{H}\}$ *in situ* NMR at high (top) and low concentration (bottom) for **step 2**, revealing the concentration dependence of the reaction.

As for any other factors as highlighted by the reviewer, solvent choice was not optimized, as tetrahydrofuran provided good solubility and compatibility with the phosphazane precursor and hydroxybenzoic acid. Templating effects were not investigated, as high selectivity could already be achieved without the necessity of additional templates.

In summary, while the procedure previously appears simple and straightforward, considerable optimization has been put into obtaining the tetrameric macrocycle **4**. While the isolated yield is moderate, it is primarily due to the high solubility of target compound preventing efficient crystallization. This is supported by the *in situ* NMR (**Figure A3**) of **step 2** reveals high selectivity for the tetramer **4** over possible side products (rearrangement products, dimers, oligomers etc.)

Figure A3. $^{31}\text{P}\{^1\text{H}\}$ in situ NMR of step 2 using the optimized reaction conditions.

The DFT calculations help to understand and justify some of the experimental results, but in some cases there is no agreement between both. The calculations have been performed only on starting materials, products and intermediates, and no transition state structure has been calculated or used in the discussion. I find this approach reasonable, considering that calculating the TSs could be problematic due to the different catalytic possibilities (surely Et₃N plays a role as a catalyst in the mechanism, but, for example, it is not easy to know a priori how many Et₃N molecules are present). However, it is important to be aware of the limitations of an approach in which the TSs are not considered at all... basically, the theoretical results could only match experimental results under thermodynamic control conditions, and it may not be the case (indeed, it is almost certainly not the case) for these reactions.

Hence, it is not possible to affirm, based on the Gibbs free energies of 2b, 2a and 2c, that the calculations support the experimental observation that only product 2a is obtained. The very similar energies of 2b, 2a and 2c would imply that a mixture of the 3 compounds will be obtained under thermodynamic control. Therefore, the only valid conclusion of the calculations is that this reaction is kinetically (not thermodynamically) controlled. The structures to the right of Figure 7 are also misleading. Compound 4-trans cannot be expected from Int I. Overall, probably an energy diagram like the one shown in Figure 7 is not the more appropriate way of showing these results, since one would expect to find in this diagram the TSs and barrier heights that could justify the reaction results under kinetic control.

ANSWER – As indicated by the reviewer, the energy diagram originally depicted (currently **Supplementary Figure S11**) could lead to misunderstandings, as it is not strictly a mechanistic study.

Attempts were made to obtain a full mechanism diagrammatic scheme (analogous to the one suggested by the reviewer including the corresponding TSs). Unfortunately, all attempts to obtain well-defined TSs for key reaction steps involving triethylamine were unsuccessful. Hence, only the energies of the intermediates were reported throughout our studies. We acknowledge that the calculated relative energies of **2a**, **2b** and **2c** does not provide a comprehensive overview of the mechanism and hence the figure has been moved to the ESI (**Supplementary Fig S11**). In fact, since all the species are relatively close in energy and only **2a** is experimentally observed, while **2b** and **2c** (which are very close in energy) are not detected throughout our studies, suggests a kinetic control of the reaction.

Overall, and for clarity, the mentioned figure has been moved to the SI, and a clarification has been made regarding the information that can be extracted from it. Moreover, the section “Mechanism of formation and computational studies” has been renamed to “Computational studies”. **Highlighted in yellow (Page 12)**

The absolute numbers of the energies of compound are also affected by the choice of the reference system. Since the reactions takes place in the presence of Et₃N, it is very likely than carboxylic acid deprotonates and forms an ionic pair with the protonated amine. Therefore, the zero-energy should correspond to this ionic pair. Solvent effects, that will be crucial in this case, would imply the use of a solvent model, and the translational entropy, that is modelled in QM packages using a gas-phase model, will also include a considerable error. Indeed, the numbers found are too large to be realistic (≈ 60 kcal/mol is similar to the heat of combustion of H₂!). However, luckily these numbers are not useful in the discussion, so I can save the effort of questioning their validity. My suggestion would be to get rid off these numbers in the discussion. In any case, they must not contain more than a single decimal figure.

ANSWER – As highlighted by the reviewer, the energies differences between the different intermediates are in some cases significantly high. This could be indeed due to the choice of the reference, as well as other factors like the mentioned translational entropy and/or the basis set superposition error (BSSE). Regarding the choice of a different energy reference, the calculation of the energy of the ionic pair Et₃N/hydroxybenzoic acid has been performed and used as zero. Unfortunately, a similarly high energy difference (in fact 10 kcal/mol higher) was observed. However, as pointed out by the reviewer, and since the values obtained do not constitute a crucial part of the discussion the scheme has been moved to the ESI (**Supplementary Figure S11**) and the discussion has been modified in the manuscript.

However, calculations are useful to justify, on the basis of the geometries of the intermediates and rotation barriers, the preference for the formation of 4 cis over 3b (at least qualitatively). In my opinion, the rotation results included in the SI section should replace the energetic diagram and the discussion should focus more on these rotational barriers than in the energies of the products and intermediates.

ANSWER – Following reviewer's 1 comments the structural features of *Int 1* leading to **4** instead of **3b** have been highlighted in the main text (in **pages 13-15**). Similarly, the qualitative analysis of the energy barriers of the rotations for **2a** and **2b** has been also included and highlighted in the main text (in **pages 15-16**).

The calculations are also useful to establish the way of association of the phosphazane macrocycle with diamines. But in this case what I miss is the experimental observation of this complex. Ideally, the complex would crystalize and the structure would be observed by X-ray diffraction experiments, or more sophisticated NMR experiments (NOESY, DOSY, etc) could confirm the association. But even a simple NMR experiment would reveal displacement of the signals of the macrocycle upon the addition of the diamine, to confirm the theoretical expectations.

ANSWER – Experimental host-guest studies were not able to be performed due to the low solubility of the bipyridinium guest. From our attempts, the bipyridinium guest was virtually insoluble in compatible anhydrous organic solvents, ranging from chloroform, acetone to acetonitrile. Small amounts of water are necessary to solubilise the guest compound in those solvents, which is chemically incompatible with **4**. This thus precludes the simple NMR experiments using the model pyridinium compound. Thus, the experimental host-guest studies remain inconclusive at this stage. However, our DFT studies indicates the reported macrocycle as good candidates for host-guest studies helping set future experimental lines of inquiry.

Reviewer #2 (Remarks to the Author):

This paper reports a new two-stage synthesis of an unfolded inorganic/organic cyclodiphosphazane macrocycle, which has been isolated and structurally characterized. The new precursors were carefully chosen to overcome the previous limitations of dimerization and allows isolation of a larger tetramer. The design features were clearly explained. Theoretical investigations provided a reasonable explanation to the observed selectivity of the macrocycle in terms of energy. Although this paper offers strong contributions in the synthetic aspect (however, reporting only one fully characterized compound), my concern is that the host-guest properties of the featured product **4** are only theoretical, which limited the impact of this paper. I have the following questions for the authors to address before I can consider recommendation for publication:

1. The capability (“outperforming traditional organic building blocks”) of most of the cyclophosphazane as building blocks cited in this paper (ref 20, 17-23) utilizes a N-H moiety for hydrogen bonding. Does the synthetic strategy offered by this paper apply to N-H derivatives? Alternatively, will the tert-butyl derivative offer superior guest-host properties in a similar way or any other way?

ANSWER – In our preliminary studies of using 4-aminobenzoic acid as organic linker, *in-situ* ³¹P NMR spectroscopy of the synthesis of the *cis* prearranged building block (**Step 1**) suggests that the reaction successfully follows the pK_a strategy proposed in this manuscript. Addition of a solution containing [CIP(μ-NtBu)]₂ (1 equiv.) and triethylamine (2 equiv.) in THF to a solution containing 4-aminobenzoic acid in THF at 0°C resulted in the exclusive formation of what we believe to be compound **A1** based on the observed chemical shift (169 ppm) in *in-situ* ³¹P NMR (see **Figure A4** below).

However, reaction of compound **A1** with an additional equivalent of [CIP(μ-NtBu)]₂ in the presence of triethylamine saw no reaction even after stirring overnight, indicative of the lower reactivity of compound **A1** *versus* that of compound **2a** presented in this manuscript. In this regard, the direct translation of the conditions employed in **step 2** is

not viable for the NH derivative (4-aminobenzoic acid) and other alternative routes needs to be explored – which will be explored in the near future.

Figure A4: In-situ ^{31}P NMR of the prearranged NH_2 derivative building block.

In terms of the bridging substituents, we have not performed any studies using alternative bridging groups. The tert-butyl group is the conventional group used in structural phosphazane chemistry. We acknowledge that in small phosphazane macrocycles, in both fully inorganic and hybrid species, replacing the *tert*-butyl group for a smaller substituent could have a major impact in the macrocycle size and host ability; however, in larger species, such as the compound herein studied, replacing the tert-butyl group for a smaller alkyl substituent will play a minor role in its host-guest properties.

2. Page 10 Line 180: 32% of Compound 4 was obtained from the first batch of crystals. What is the total isolated yield of 4? If it's not over 60-70% at least, this reaction is not as selective as it was claimed.

ANSWER – The high selectivity for compound 4 is supported from our *in situ* NMR obtained during step 2 (Figure A3, *vide supra*). As seen from the spectrum, high conversion was observed, compared to the small amounts side reactions. However, compound 4 is highly soluble, preventing efficient crystallization, resulting in the moderate isolated yield. The terminology first batch of crystals has been changes to total listed yield throughout the manuscript to avoid confusion.

3. Page 11 Line 199-200: Is there any implications of this unprecedented rotation of the central P(III)₂N₂ units on its host-guest properties?

ANSWER – The observed “swivel” movement of the central P^{III}₂N₂ unit suggests the presence of an “adaptative” cavity capable of a geometrical change from cylindrical to conical, which in turn could benefit potential host-guest interactions.

4. How is the thermal stability and solubility of the compound 4? The authors mentioned the air- and moisture sensitive nature which prevents the host-guest studies. What about simple NMR experiments? If compound 4 is not thermally stable, what does it decompose into?

ANSWER – Compound 4 is thermally stable under moisture-free and inert atmosphere. It can be heated in both hexanes and toluene, while solid samples can be heated to 180°C in melting point experiments. The comment on moisture sensitivity preventing experimental host-guest studies is due to the low solubility of the bipyridinium guest. From our attempts, the bipyridinium guest was virtually insoluble in compatible anhydrous organic solvents, ranging from chloroform, acetone to acetonitrile. Small amounts of water are necessary to solubilize the guest compound in those solvents,

which is incompatible with **4**. This thus precludes the simple NMR experiments using the model pyridinium compound.

5. For a synthesis-focused paper, there is only one fully characterized compound (**4**). Although NMR assignment has been shown in the current version of SI, I don't see experimental evidence for the reported assignment. The 2D NMR evidence for the assignment should be added in the SI. In figure S5, is the carbon signal assigned to **9** because it showed coupling to P?

ANSWER – The carbon NMR for compound **4** was assigned based on known assignments of the organic linker, 4-hydroxybenzoic acid. The reaction resulted in minimal deviations in the carbon NMR chemical shifts, and the assignment of compound **4**'s spectrum was based on the linker. (Figure A5)

Figure A5. Comparison of the aromatic region in the ¹³C NMR with 4-hydroxybenzoic acid

Reviewer #3 (Remarks to the Author):

In this work, the innovative approach which is alternative to already conventional synthetic route has been developed, allowing the tetrameric macrocycles. This approach has an importance for supramolecular chemistry field in terms of the control of macrocyclic size. The paper which is written by Garcia et al. is original and acceptable for publishing in Communications Chemistry after minor revisions have been implemented. Some recommendations are given for authors to enhance the quality of the paper below:

1. The energy levels of compound 2a and 2b are very close to each other to synthesize compound 4 (cis). In addition, the difference between their values and compound 2c is nearly 0.49 kcal/mol. So, both compound 2a and 2b are possible candidate for the synthesis of compound 4. Only the chemical shift value in ^{31}P NMR spectrum cannot be distinctive method. The reason why compound 2a or 2b is starting compound for the synthesis of compound 4 must be clarified.

ANSWER – In addition to the relative energies between **2a** and **2b** discussed for the issues raised by Reviewer 1, the ^{31}P NMR chemical shift differences between **2a** and **2b** is expected to be very distinct, with ~170 ppm expected for **2a** and ~140 ppm for **2b**. Furthermore, the reaction of **1** with 2 equivalents of 4-cyanophenol or 2 equivalents of 4-cyanobenzoic acid resulted in the disubstituted phosphazanes, with 144 and 173 ppm respectively. See: Inorg. Chem., 2015, 54, 13, 6423–6432 and Chem. Eur. J. 2017, 23, 11279 – 11285.

As such, we believe that ^{31}P NMR is sufficient to unambiguously differentiate between the two isomers due to the distinctive chemical shift differences between the P-O-R and P-OC(O)-R moieties.”

2. The ^{31}P NMR spectrum of the reaction mixture in which newly synthesized compound 4a were obtained, should be given in the paper. It was mentioned that compound 4 was obtained in a yield of 32%. What about the remaining part (68%)? Do the new compounds form or compound 2a remains without reacting? This should be clarified.

ANSWER – As already answered for Reviewer 2, the *in situ* NMR (**Figure A3**) shows that the major product of the reaction is compound 4, with high selectivity. However, the high solubility of compound 4 prevents efficient recrystallization resulting in low isolated yield during crystallisation.

3. In Supplementary file, “ first batch of crystals” was written for compound 4 in the experimental section. Why does it mention as first batch of crystals???

ANSWER – As already answered to Reviewers 1 and 2, the high selectivity for compound 4 is supported from our *in situ* NMR obtained during **step 2 (Figure A2-A3)**. Our *in situ* NMR studies demonstrate high conversion with a relatively small amounts of products. However, compound 4 is highly soluble preventing efficient crystallization, which results in the moderate isolated yields. The terminology first batch of crystals has been changes to total listed yield throughout the manuscript to avoid confusion.

4. Is it possible to get different result if different base (for ex. NaH) rather than triethylamine was used? Why triethylamine was preferred?

ANSWER – In general, nucleophilic bases have been reported to react with the starting material to $[\text{CIP}(\mu\text{-NtBu})_2]$ substituting the chlorides present. Thus, it is common to employ non-nucleophilic bases to prevent side reaction. Generally, triethylamine is used due to its low cost, although other non-nucleophilic bases will work as well. Therefore, triethylamine was used throughout our studies.

5. There are some mistakes in the paper:

Page 10 line 168: It was said trans dimeric macrocycle (3a) was not observed but in Figure S2, it is seen it formed even little amount and marked with symbol at about 140 and 150 ppm.

ANSWER – the sentence has been changed to “Additionally, no signals for the *trans* di-substituted (2c, in **Figure 4**) monomer (*i.e.*, mixed aryl oxide and alkoxide), and negligible amounts of the previously reported *trans* dimeric macrocycle (3a, **Figure 4 bottom**) were observed (**Supplementary Figure S2**).” **Highlighted in the manuscript (Page 10)**

Page 10 line 176: ‘Resonance signals at 180 and 150.1 ppm are not compatible with the data in Supplementary file (Figure S3). The values should be corrected.

ANSWER – The manuscript has been corrected to “Resonance signals at δ 181.3 and 152.4 ppm” **Highlighted in yellow (Page 11)**

Page 10 line 178: The in situ $^{31}\text{P}\{^1\text{H}\}$ NMR spectra recorded revealed two singlet 175 resonance signals at δ 180.0 and 150.1 ppm along with small resonance signals corresponding to the dimeric trans macrocycle 3a, presumably due to fragmentation and rearrangement products of 2a or its derivatives (see Supplementary figures S2-S6)... Figure S5 is related to ^{13}C NMR spectrum of compound 4, so this is not relevant.

ANSWER – the text has been modified to “presumably due to fragmentation and rearrangement products of 2a or its derivatives (for $^{31}\text{P}\{^1\text{H}\}$, ^1H and ^{13}C NMR spectra of 4 see **Supplementary figures S2-S5**, respectively).^{42,59}” **Highlighted in yellow (page 11)**

6. It is seen the R factors are high (R1, wR2 ($I > 2\sigma(I)$) 0.1331, 0.3575 R1, wR2 (all data) 0.1588, 0.3854), which means there is a problem in crystal quality but $I/\sigma(I)$: 20.6 Rint

factor below 10%. Why does this contradiction arise? The crystal data were collected at 100 K, but I see they were collected at 296 K / 23 °C. Low temperature data collection may prevent rotation of the phenyl ring and tert-butyl group. The data should be collected again at low temperature.

ANSWER – The crystal data were collected at 100 K. This is reported both in the CIF and the SI.

7. The capped stick model is preferred as the crystal displaying in both the paper and the supplementary part. ORTEP drawings should be given by showing the atom-labelling scheme. The bond parameter is included in Figure S10, but these parameters create uncertainty since the atoms were not labelled.

ANSWER – An ORTEP diagram is provided in the Checkcif and the labelling scheme is reported in the CIF – because of the disorder visualising the bond labelling scheme is best done with software – we recommend Mercury. We have removed the bond lengths from the Figure S10 caption as all the bond length data is in the cif.

8. You need to update your cif. file by labelling the atoms correctly. The alert A and B errors on the checkcif file should be explained.

ANSWER – The structure has been relabelled. The Alert A and B errors are due to the disorder present in the structure and have been addressed in the crystallography section of the SI.

REVIEWERS' COMMENTS:

Reviewer #1 (Remarks to the Author):

After reading the response from the authors and the new version of the manuscript, I confirm that the modifications increase the quality of the manuscript to a point that it can now be accepted in Communications Chemistry.

The manuscript describes the novel (and clever) preparation of a macrocyclic phosphazane compound. The work is convincing, the literature references are appropriate, the manuscript includes the details required for reproducing results, and the conclusions are reasonable.

Luis Simón

Reviewer #2 (Remarks to the Author):

I'm satisfied with the response the authors provided to the referees' comments and the corrections on the MS and SI. I recommend publication of the revised MS at Communications Chemistry.

Reviewer #3 (Remarks to the Author):

The corrections were made after this revision, so the paper can be acceptable for publishing Communications Chemistry.

We thank the three reviewers for their time in reviewing our manuscript and their valuable and positive feedback.

Reviewer #1 (Remarks to the Author):

After reading the response from the authors and the new version of the manuscript, I confirm that the modifications increase the quality of the manuscript to a point that it can now be accepted in Communications Chemistry.

The manuscript describes the novel (and clever) preparation of a macrocyclic phosphazane compound. The work is convincing, the literature references are appropriate, the manuscript includes the details required for reproducing results, and the conclusions are reasonable.

Luis Simón

Reviewer #2 (Remarks to the Author):

I'm satisfied with the response the authors provided to the referees' comments and the corrections on the MS and SI. I recommend publication of the revised MS at Communications Chemistry.

Reviewer #3 (Remarks to the Author):

The corrections were made after this revision, so the paper can be acceptable for publishing Communications Chemistry.